# Prebiotics, Probiotics, and Postbiotics in the Prevention and Treatment of Anemia

**DOI:** 10.3390/microorganisms10071330

**Published:** 2022-06-30

**Authors:** Zuzanna Zakrzewska, Aleksandra Zawartka, Magdalena Schab, Adrian Martyniak, Szymon Skoczeń, Przemysław J. Tomasik, Andrzej Wędrychowicz

**Affiliations:** 1Department of Pediatric Oncology and Hematology, Institute of Pediatrics, Jagiellonian University Medical College, 30-663 Krakow, Poland; zuzanna.zakrzewska93@gmail.com (Z.Z.); magdalenaschab95@gmail.com (M.S.); szymon.skoczen@uj.edu.pl (S.S.); 2Department of Pediatrics, Gastroenterology and Nutrition, Institute of Pediatrics, Jagiellonian University Medical College, 30-663 Krakow, Poland; azawartka@usdk.pl (A.Z.); andrzej.wedrychowicz@uj.edu.pl (A.W.); 3Department of Clinical Biochemistry, Institute of Pediatrics, Jagiellonian University Medical College, 30-663 Krakow, Poland; adrian.martyniak@uj.edu.pl

**Keywords:** iron deficiency, iron absorption, lactobacillus, bifidobacterial, FOS, GOS, fiber, SCFA, tryptophan

## Abstract

Iron deficiency anemia (IDA) is very common and affects approximately 1/3 of the world’s human population. There are strong research data that some probiotics, such as *Lactobacillus acidophilus* and *Bifidobacterium longum* improve iron absorption and influence the course of anemia. Furthermore, prebiotics, including galactooligosaccharides (GOS) and fructooligosaccharides (FOS), increase iron bioavailability and decrease its destructive effect on the intestinal microbiota. In addition, multiple postbiotics, which are probiotic metabolites, including vitamins, short-chain fatty acids (SCFA), and tryptophan, are involved in the regulation of intestinal absorption and may influence iron status in humans. This review presents the actual data from research studies on the influence of probiotics, prebiotics, and postbiotics on the prevention and therapy of IDA and the latest findings regarding their mechanisms of action. A comparison of the latest research data and theories regarding the role of pre-, post-, and probiotics and the mechanism of their action in anemias is also presented and discussed.

## 1. Introduction

Anemia is a disease in which the level of the number of red blood cells or the hemoglobin concentration within them is below the reference range. Anemias can affect around 1/3 of the world’s human population [1,2]. The most common is iron deficiency anemia. It can affect women and elderly people, in addition to children in developed and developing countries [3,4,5]. It can lead to different less severe or more severe consequences. Iron plays a crucial role in the human metabolic process with respect to oxygen transport, electron transfer, oxidation-reduction reactions, hormone synthesis, and nitrogen fixation [6,7]. Disturbances in the adequate iron level in an organism can lead to impairment of body functions [8]. In the world, in 2016, there were an estimated 24,000 deaths from iron deficiency anemia [7]. Causes of iron deficiency can be related to diet (lack of iron-rich food), the inability to properly absorb iron, chronic loss of blood (e.g., from the GI tract) and/or high requirements during different periods of age. Iron can be taken from food, such as meat or vegetables, medications or transfusions, but the source of iron is crucial, as well as the type of iron [8,9]. Several mechanisms that help to maintain adequate iron absorption and tissue level are related to symbiotic bacteria in the gut. The most common and well-recognized is fermentation, which improves the absorption of this element [3,8,10].

### 1.1. Prebiotics

The current definition of prebiotics according to the International Scientific Association for Probiotics and Prebiotics (ISAPP) is ‘a substrate that is selectively used by host microorganisms that confer a health benefit’ [11]. Although most commonly prebiotics are carbohydrate based, other substances, such as polyphenols and polyunsaturated fatty acids converted to their respective conjugated fatty acids are also utilized by the microbiota and, therefore, could be included in the definition. Over the past 20 years, the scope of the term ‘prebiotics’ has changed due to the development of research tools (such as next-generation sequencing), the expansion of knowledge of the microbiota and the multidirectional interactions between the diet, microorganisms, and host system. Prebiotics are usually administered orally, but they can be given directly to the intestines or other sides of the body, such as the skin.

Currently, to be qualified for this group, the product must be resistant to the hydrolysis process, acidic pH of the stomach, mammalian gastrointestinal enzymes and must reach the colon in the undigested form where it is fermented by bacteria [12,13]. Moreover, it has to decrease the pH of the intestinal content and stimulate the growth and activity of bacterial species with a positive impact on host health [12,13]. The most well-known prebiotics should include fructooligosaccharides (FOS), galactooligosaccharides (GOS), inulin, glucans, pectins and lactulose [12]. Prebiotics are naturally found in food products, such as asparagus, garlic, onion, wheat, honey, cow’s milk, and bananas and can also be produced industrially [11,12]. There are studies in the literature on the use of nanoparticles in the prebiotic industry. They indicate that the use of nanotechnology enables the new use of prebiotics in the form of nanoprebiotics through properties, such as improving drug delivery and antimicrobial effects [14].

### 1.2. Probiotics

The human gastrointestinal tract is home to trillions of different microorganisms that have a multidirectional impact on host metabolism and the functions of multiple organs. All microbes that live in the gut create a complex ecosystem called the human microbiota. Many recent metagenomics studies have shown that the intestinal microbiota of each person consists of a unique collection of microorganisms: approximately 150 to 400 bacterial phyla, accompanied by fungi, viruses, and archaea. The predominant bacterial phyla Bacteroidetes, Firmicutes, Proteobacteria, and Actinobacteria comprise 95% of the human microbiota. The specific composition of the individual microbiome varies extensively and depends on many factors, such as host genetic variation, mode of delivery, type of newborn feeding, age, diet, lifestyle, diseases, and medications [15]. Probiotics are living microorganisms that have several important functions that have a positive influence on human health [16]. Among several groups of bacteria with probiotic action, the largest is lactic acid producing bacteria (LAB) [17,18,19]. The billions of probiotic bacteria are vulnerable to factors, such as the overuse of antibiotics and food preservatives. Probiotics must be introduced to the organism by food as well as by diet supplements and their meaning for the human organism depends on the properly estimated dose, which is crucial to achieving a positive effect [17,20]. There are several different mechanisms by which probiotics work [17,21]. They can modulate the microbiota of the intestinal tract and have properties to interact with the immune system. In clinical research, probiotic bacteria, such as *Lactobacillus acidophilus* or *Bifidobacterium longum* have anti-inflammatory properties [20]. One of the most important known actions of probiotics is to play a dominant role over pathogenic microorganisms. In addition, they help in the production of short-chain fatty acids (SCFA), vitamins, bactericidins, and the metabolism of bile acid salts [17,18,19]. Some of these bacteria and their metabolites are important in iron absorption and influence the course of anemia. However, the final effect of probiotics is strain-specific and dose-dependent [16].

### 1.3. Postbiotics

The term postbiotics, sometimes also called metabiotics, refers to the structural components of probiotic microorganisms, their metabolites, and signaling molecules with a determined chemical structure that can optimize host-specific physiological functions and regulatory, metabolic, and/or behavior reactions related to the activity of the host’s indigenous microbiota [22,23]. According to a new definition (2021) of postbiotics prepared by The International Scientific Association of Probiotics and Prebiotics (ISAPP), postbiotics are “a preparation of inanimate microorganisms and/or their components that confers a health benefit on the host” [22]. This definition focuses mostly on nonviable microbes and their cell components as active substances. Metabolites, according to this definition, should be considered postbiotics only in the connection and presence of non-viable microbes or their parts. Postbiotics are found in all natural fermented products, such as kefir, kimchi, sauerkraut, tempeh, yogurt, and certain pickles, as well as inside the human gut. The most important postbiotics metabolites are vitamins, organic acids, short chain fatty acids, and amino acids, such as tryptophan (Trp). Depending on the type of microorganism, the strain, and the metabolism product, the effects of postbiotics are very different, as is the case of prebiotics. The benefits of postbiotics can be direct or indirect [24]. Direct benefits result from the action of postbiotics on host cells. Indirect benefits include the creation of an acidic environment in the intestinal tract or the formation of iron-chelating ligands by which iron absorption increases [12,25,26].

The paper summarizes the results of research on the influence of prebiotics, probiotic bacteria and their metabolites (postbiotics) on the prevention and treatment of anemia as well as the mechanism of their action on the body’s iron status. In this review, we will focus mainly on iron deficiency anemia (IDA), since it is the most common in the human population in the world, and this topic has been researched primarily by scientists.

## 2. Prebiotics in Anemia

Galactooligosaccharides (GOS) are a group of carbohydrate polymers connected by β(13), β(14) and β(16) linkages, which contain galactose in their structure [12]. The natural source of GOS is human milk and cow milk. GOS are soluble in water; their stability in an acidic environment and high temperature depends on the length of the chain and the distribution of bonds [27]. They are formed as a result of the transformation of lactose into glycosidic hydrolases, the most popular of which is b-galactosidase [28]. GOS is believed to increase iron absorption [12]. This may be due to the ability of prebiotics to lower the pH of the colon, increasing the reduction of Fe^3+^ to Fe^2+^, as well as the fermentation process that enhances SCFA production, which may contribute to increased colon absorption area [29]. In a study by Pagini et al., it was shown that Kenyan infants fed for 3 weeks with corn porridge enriched in ferrous fumarate (FeFum) and ferric sodium EDTA and 7.5 g of GOS, had a 62% increased iron absorption compared to the group not receiving GOS [30] (Table 1). However, another study showed that in infants with anemia or iron deficiency, a single dose of 7.5 g GOS added to a meal did not significantly increase iron absorption [31] (Table 1). Christides et al., compared the bioavailability of iron from infant formulas, which differed in the content of prebiotics and ascorbic acid [32]. It was shown that milk with the highest content of prebiotics and ascorbic acid had the highest bioavailability of iron. Importantly, after equalizing the amount of prebiotics in the tested infant formula, no difference in the bioavailability of iron was observed [32]. It is also known that iron supplementation can have gastrointestinal side effects. Therefore, it is worth noting that the addition of GOS during iron supplementation reduces the destructive effects on the intestinal microbiota [33]. It was shown that infants supplemented with iron and GOS by 4 months, had more Bifidobacterium and Lactobacillus bacteria and a smaller amount of Clostridiaes compared to the group receiving only iron supplements. Moreover, no significant differences were found in the number of individual bacterial strains between the iron with GOS group and the control group not receiving Fe. It was also noticed that the level of intestinal fatty acid binding protein was higher in the group of infants receiving Fe supplementation compared to the Fe + GOS group, but it did not differ significantly between the GOS group and the control group [33]. In adults with anemia, studies also confirm the beneficial effects of GOS on iron absorption. Jeroense F. et al., showed that in women with iron deficiency, the addition of GOS increased fractional iron absorption (FIA) from FeFum by 61% after a single administration with water, and by 28% after administration with a meal [34] (Table 1). Researchers also checked whether long-time supplementation with GOS would improve iron absorption. It was shown that after 15 g GOS per day supplementation for four weeks, iron absorption from FeFum in a tested meal without GOS did not change. Moreover, GOS did not improve iron absorption from FeSO_4_, which was also confirmed by other authors [34,35] (Table 1).

Fructooligosaccharides (FOS) are a group of oligosaccharides fermented in the distal part of the colon [36]. They are found naturally in onions, garlic, wheat, asparagus, and bananas [37]. FOS can be obtained from sucrose by the transfructosylation reaction with the use of the enzyme β-fructofuranosidase, or by inulin hydrolysis [38]. The role of FOS in iron absorption in humans remains unclear [36]. The more profound studies were performed on rats. Lobo et al., investigated the effect of FOS on the bioavailability of iron in anemic rats [39]. They showed that the addition of FOS improved the bioavailability of iron, which was measured by the level of hemoglobin and iron stores in the liver [39] (Table 2). In another study, also in rats with iron deficiency anemia, it was shown that the addition of FOS in the group of rats fed an iron-deficient diet slightly improved bioavailability compared to the group without FOS. On the other hand, in the group fed a diet with proper iron content, the addition of FOS did not change the bioavailability of iron [40]. Moreover, Ohta et al., showed that FOS prevented anemia in totally gastrectomized rats [41] (Table 2). Sakai et al., observed the effect of short-chain FOS and inulin on the development of anemia in gastrectomy rats [42] (Table 2). It was shown that the values of hemoglobin and hematocrit were significantly higher in the group supplemented with FOS compared to the control group and the group supplemented with inulin. The same authors investigated whether the mechanism of increasing iron absorption by FOS was related to the absorption in the large intestine. For this purpose, a group of rats was divided into subgroups after gastric removal, after removal of the cecum, or after removal of both the stomach and the cecum, and half of each group was fed with an FOS diet. It was observed that the addition of a prebiotic significantly prevented the decrease in hematocrit and hemoglobin, but in the group of rats, after both the stomach and cecum were removed, the ability to prevent anemia by FOS was significantly reduced. This proves that in gastrectomy rats, FOS prevents the occurrence of anemia by acting in the large intestine [43].

Inulin is another type of prebiotic that belongs to the group of fructans together with FOS [44]. Food sources include artichokes, asparagus, bananas, onions, garlic, wheat, chicory, and mushrooms. Inulin supplementation improves stool consistency after radiotherapy and the increase in the amount of butyrate [45]. In a study by Castro et al., it was observed that in children who were supplemented with iron, zinc, copper, vitamin A, vitamin C and inulin for 45 days, five times a week, the values of hemoglobin, hematocrit, MCV, MCH and ferritin improved. The authors suggest that one of the reasons may be the addition of inulin [46]. However, in the case of anemia, not all studies confirm the role of inulin in increasing iron absorption. Petry et al., showed that supplementation of 20 g of inulin three times a day for 4 weeks to anemic women did not increase iron absorption [47] (Table 1). The effect of inulin on iron absorption has also been investigated in animals. A study by Mohammed et al., showed that the addition of inulin to yogurt with Fe_2_(SO_4_)_3_ administered to rats increased the bioavailability of iron, and the greatest effect was observed in the case of long-chain inulin [48] (Table 2). The effect of increased iron absorption in the intestine after inulin ingestion in rats was also shown by Freitas et al. [49] (Table 2). However, in a study by Zhang et al., it was not observed that inulin increased iron absorption in anemic rats [50] (Table 2).

Pectins are one of the most common plant cell-wall polysaccharides [51]. It is a type of water-soluble fiber, and its structure contains galacturonic acid [52]. The types of pectins differ in the degree of esterification. In food, it is found in citrus, apples, plums and pears, as well as in vegetables, legumes and nuts [51]. There are only a few studies assessing the effect of pectin on iron absorption. Jaramillo et al., assessed the effect of 5 g of pectin with various degrees of esterification on iron absorption with 5 mg of FeSO_4_ in healthy women. There were no significant differences depending on the degree of pectin esterification [53]. In another study, the same author showed no effect of pectin in a dose of 250 mg on the bioavailability of iron with FeSO_4_ in the fasting women [54]. In a study by Bosscher et al., it was observed that the addition of pectins to infant formula reduces the bioavailability of iron, and this effect is greater in the case of pectin with a low degree of esterification compared to high esterified pectin [55]. Additionally, Kim et al., showed the effect of pectins on the bioavailability of iron. They observed that in the group of rats with anemia, the addition of high-esterification and low molecular weight pectins to the diet was associated with higher hemoglobin regeneration efficiency, hematocrit values, serum iron level and transferrin saturation compared to the control [56] (Table 2).

**Table 1 microorganisms-10-01330-t001:** Summary of human studies on prebiotics in anemia.

Prebiotic/ Author	Dose	Type of Administration	Study Group (*n*)/Control Group (*n*)	Assessment Method	Main Result
GOS; Paganini, D. et al., (2017) [30]	7.5 g GOS for 3 weeks	fortified maize porridge	healthy infants fed GOS diet (*n* = 28)/no GOS diet (*n* = 22)	erythrocyte incorporation of stable isotopes	the relative iron bioavailability was significantly increased in study group (*p* = 0.006) GOS increased iron absorption from FeFum (*p* = 0.047) but not from FeSO_4_ (*p* = 0.653)
GOS; Mikulic, N. et al., (2021) [31]	7.5 g GOS added to a single test meal	fortified maize porridge	iron deficient and anemic infants fed GOS diet (*n* = 12)/no GOS diet (*n* = 11)	fractional iron absorption (FIA) assessed by erythrocyte incorporation of isotopic labels	GOS added to a single iron-fortified test meal did not significantly increase iron absorption
GOS; Jeroense, F.M.D. et al., (2019) [34]	15 g GOS for 4 weeks	with water or with meal	iron-depleted women (*n* = 34)	FIA assessed as erythrocyte incorporation of stable isotopes	GOS given with FeFum significantly increased iron absorption when was consumed with water (+61%) and with a meal (+28%); 4 weeks of GOS consumption was associated with an increase in hemoglobin (Hb) level (*p* = 0.001)
GOS; Jeroense, F.M.D. et al., (2020) [35]	3.5 g, 7 g, 15 g GOS	with water	iron-depleted women assigned to groups with different conditions (*n* = 46)	FIA assessed as erythrocyte incorporation of stable isotopes	7 g GOS significantly increased FIA from FeFum (+26%; *p* = 0.039) 3.5 g GOS did not significantly increased FIA from FeFum (*p* = 0.130) 15 g GOS did not significantly increased FIA from FeSO_4_ (*p* = 0.998) or FePP (*p* = 0.059) AA given with FeFum and 7 g GOS significantly increased FIA compared with FIA from FeFum given with 7 g GOS alone (+30%; *p* < 0.001)
Inulin; Petry N. et al., (2012) [47]	20 g/d for 4 weeks	fibruline instant dissolved in water	women with low iron status (*n* = 32)	FIA was assessed by using stable-iron-isotope techniques	no significant differences (*p* = 0.10) between mean FIA in the inulin and placebo

AA—Ascorbic Acid AA, FeFum—ferrous fumarate, FIA—fractional iron absorption, Hb—hemoglobin concentration, FeSO_4_—ferrous sulfate, FePP—ferric pyrophosphate.

**Table 2 microorganisms-10-01330-t002:** Summary of animal studies on prebiotics in anemia.

Prebiotic/ Author	Dose	Type of Administration	Subject Study Group (*n*)/Control Group (*n*)	Assessment Method	Main Result
FOS; Lobo AR. et al., (2014) [39]	7.5% FOS for 1 or 2 weeks	yacon flour or Raftilose P95	iron deficient anemic rats supplemented with FP assigned to RAF group (*n* = 16) or YF (*n* = 16)/control group (*n* = 16)	HRE, hepatic Fe stores	FOS supplementation increased Fe bioavailability measured by HRE and hepatic Fe stores, which were more pronounced in the RAF group at week 1 changes in Hb level in FOS-fed rats were greater than those in the FP group (*p* = 0.01) and similar to those in the FS group
FOS, Ohta A. et al., (1998) [41]	75 g/kg for 6 weeks	added to diet	rats after surgically stomach removing (*n* = 7) or sham operated rats (*n* = 7) fed the FOS diet/no FOS diet (*n* = 7)	Ht, Hb concentration, HRE	Ht, Hb concentration, and HRE were significantly lower in gastrectomized rats fed a diet without FOS compared to the other three groups FOS prevented anemia in totally gastrectomized rats
FOS; Zhang F. et al., (2017) [40]	5% (*w*/*v*) FOS = 1–2/g per day for 28 days	dissolved in water	non anemic rats fed regular diet + FOS (*n* = 6)/regular no FOS diet (*n* = 6) anemic rats fed regular diet + FOS (*n* = 6)/regular no FOS diet (*n* = 6) anemic rats fed low iron diet + FOS (*n* = 6)/low-iron no FOS diet (6)	Hb concentration	anemic rats fed low-iron diet + FOS had higher Hb level (*p* < 0.05) after 21 days, compared to control group in anemic rats fed regular diet, Hb returned to normal level after 14 days and FOS supplementation showed no additional effects
Sc-FOS, inulin; Sakai K. et al., (2000) [42]	Sc-FOS (75 g/kg diet) or inulin (75/kg diet) for 6 weeks	with diet	gastrectomized rats fed sc-FOS diet (*n* = 5) or inulin diet (*n* = 5)/gastrectomized (*n* = 5) or sham operated (*n* = 5) rats fed control diet	Hb concentration, Ht, HRE	in gastrectomized rats Hb and Ht levels were significantly higher in the group fed the Sc-FOS-containing diet compared with levels in rats fed the control diet or the inulin-containing diet HRE after 3 weeks in the FOS diet group was significantly higher than HRE in the other groups
Sc-FOS; Sakai K. et al., (2000) [43]	Sc-FOS 75 g/kg diet for 28 days	with diet	sham-operated (*n* = 7)/GX (*n* = 7)/CX (*n* = 7)/GCX (*n* = 7) rats fed Sc-FOS diet (*n* = 7)/control diet (*n* = 7)	Ht, Hb and SI concentration, UIBC, Hb-Fe, HRE	Hb and Ht in the GX rats without cecectomy and fed the Sc-FOS diet were higher than those in the control group (*p* < 0.05) GX rats without cecectomy and fed the Sc-FOS diet had significantly higher UIBC and HRE compared with gastrectomized rats without cecectomy and fed the control diet the effectiveness of Sc-FOS in preventing postgastrectomy anemia was significantly decreased by cecectomy
Inulin; Mohammad O. et al., (2021) [48]	4% long or short-chain inulin by 4 weeks	in yogurt	anemic rats fed inulin free yogurt (*n* = 8) or yogurt containing long-chain inulin (*n* = 8) or short chain inulin (*n* = 8)/anemic (*n* = 8) or non-anemic (*n* = 8) rats fed inulin-free yogurt (*n* = 8)	Hb concentration, RBC, Ht, SI content	long-chain inulin exhibited the best effects in terms of iron supplementation and bioavailability
Inulin; Freitas K.C. et al., (2012) [49]	100 g/kg of ration by 21 days	with diet	anemic rats fed HP inulin (*n* = 12) or HP inulin + oligofructose (*n* = 12) or oligofructose (*n* = 11)/control diet (*n* = 12)	intestinal absorption of Fe, Hb concentration	HP inulin and oligofructose increased the intestinal absorption of Fe in rats values of Hb in the HP inulin were significant higher (*p* ≤ 0.001) than in the control group
inulin, FOS, GOS; Zhang F. et al., (2021) [50]	5% (m/V) (1–1.5 g/d) prebiotic by 35 dys	dissolved in water	45 u Fe/g diet (*n* = 8) or 12 u Fe/g diet (*n* = 8) + short-chain FOS + long-chain inulin (*n* = 8)/45 u Fe/g diet (*n* = 8) or 12 u Fe/g diet (*n* = 8) 12 u Fe/g diet + FOS (*n* = 8) or inulin (*n* = 8) or GOS (*n* = 8) or lactulose (*n* = 8)/12 u Fe/g diet (*n* = 8)	Hb concentration, tissue non-heme iron levels	Hb concentration in rats supplemented with GOS after 3 weeks was significantly higher than in rats without supplementation Hb concentration in rats supplemented with FOS after 4 weeks was significantly higher than in rats without supplementation inulin, lactulose, short chain FOS + long chain inulin showed no effect; prebiotics had no effects on rats with normal iron status
Pectin; Kim M. et al., (1992) [56]	80 g/kg diet	added to diet	group 1—pectin high DE and high MW (*n* = 6)/rats fed control diet (*n* = 6) group 2—pectin with high DE and low MW (*n* = 6)/rats fed control diet(*n* = 6) group 3—pectin with low DE and high MW (*n* = 6)/rats fed control diet (*n* = 6) group 4—pectin with low DE and low MW (*n* = 6)/rats fed control diet (*n* = 6) + pair fed rats: each rat was fed an amount of the control diet equal to the average consumed by its respective pectin-fed group on the previous day	HRE, MCHC, UIBC, TIBC, Ht, SI concentration	pectin did not reduce iron bioavailability rats from group 2 had higher (*p* < 0.05) HRE, Ht, SI concentration, transferrin saturation, and lower UIBC and TIBC compared with the control group rats from groups 1 and 4 had improved hematological indices compared with group 3 and the control group

FIA—fractional iron absorption, FS—ferrous sulphate, FP—ferric pyrophosphate, Ht—hematocrit, Hb—hemoglobin concentration, HRE—hemoglobin regeneration efficiency, SI -serum iron, UIBC—unsaturated iron-binding capacity, Hb-Fe—Hemoglobin-iron, Ascorbic Acid (AA), RAF—Raftilose, YF—yacon flour, GX—gastrectomy, CX—cecectomy, GCX—gastrectomy and cecectomy, MCHC—mean corpuscular hemoglobin concentration, DE—degree of esterification, HP inulin—high performance, MW—molecular weight, MCHC—mean corpuscular hemoglobin concentration, TIBC—total iron-binding capacity.

## 3. Probiotics in Anemia

A lot of studies showed a correlation between gut microbiota and iron deficiency. It is well known that there is an effect of iron deficiency on the composition of the gut microbiota and susceptibility to intestinal infections [7,57]. Additionally, some studies recording positive effects on iron absorption due to probiotic bacteria were published. Vonderheid et al., showed a positive effect of *Lactiplantibacillus plantarum 299v* on the prevention of iron deficiency anemia. This strain improves the absorption of non-heme iron in Caucasian Europeans [3,7]. Korcok et al., describe the additive effect of using *L. plantarum 299v* together with sucrosomal iron and vitamin C in the prevention and treatment of iron deficiency. In this research, one group received only vitamin C and iron, and the second was given additional *L. plantarum*. At the end of the study, the iron blood levels were higher in the second group because of increased iron absorption [25,58]. *L. plantarum 299v* was taken under the consideration in several studies focused on different types of iron deficiencies. Adiki et al., showed that the use of *L. plantarum 299v* together with pear millet enhances the absorption of iron. They found out that an increase in probiotic dose does not have a meaningful influence on hemoglobin and hematocrit [59]. Hoppe et al., reported that in people with an increased need for iron supplementation of *L. plantarum 299v* together with a meal enhances the bioavailability of iron [60].

Iron deficiency studies focused also on other potentially beneficial types of probiotics as *Limosilactobacillus fermentum* or *Lactobacillus acidophilus*. Garces et al., proved that *L. fermentum* internalizes into enterocytes delivering iron-oxide nanoparticles by which adequate levels of iron are obtained. The cooperation of these two is a new approach to increasing iron absorption [61]. In addition, *Lactobacillus acidophilus* use in iron deficiency anemia according to Khodaii et al., results in an increase in ferritin formation in the intestinal cell [62].

Vonderheid et al., analyzed the possible mechanism of probiotic strains on iron status and its prevention of iron deficiency. *L. plantarum 299v* in contrary to the use of other probiotics, enhances the process of iron absorption significantly. This process is possible due to immunomodulation, anti-inflammatory response, the formation of bioavailable ferrous form by reduction of ferric iron, and enhancing iron uptake by enterocytes [3].

But not all studies reported positive effects of probiotics on iron status. Rosen et al., used *L. plantarum 299v* in the pediatric population and found no significant difference in ferritin levels between children with iron deficiency anemia, treated with iron supplements alone, and those treated with probiotics [63]. The summarized studies with probiotics are listed in Table 3.

## 4. Metabolic Postbiotics in Anemia

### 4.1. Vitamins

The important vitamin in the course of anemia and which can also be classified as postbiotic is folic acid (FA). Folate plays an important role in the synthesis, methylation, and repair of DNA and is also known to be an antioxidant. FA is a coenzyme in the synthesis of amino acids, purines, and thymine. FA deficiency causes growth failure and anemia. In food, we can find FA in dark green vegetables, beef, eggs, and whole grains, and FA can also be synthesized by intestinal bacteria. In humans, the daily requirement for FA is covered entirely by diet or supplementation [63]. Many species of bacteria, including those living in the intestines, have the ability to synthesize FA de novo, such as *E. coli*, *L. acidophilus*, and also Bifidobacteria. Strains of high-level synthesis are *Bifidobacterium bifidum* and *Bifidobacterium longum* subsp. *infantis* and low-level folate-producing species are *Bifidobacterium breve*, *Bifidobacterium longum* subsp. *longum* and *Bifidobacterium adolescentis* [65]. The study by Strozzi et al., confirmed a higher fecal FA level in patients colonized by the three *Bifidobacterium* spp. groups (*p* = 0.004 in group A, *p* < 0.001 in group B and *p* = 0.049 in group C) [66]. Lactobacilli are another common group of human intestinal commensals and have recently been researched as potential producers of folic acid. *Lactobacilus* spp. has been successfully tested for the production of folate-enriched fermented dairy products [67]. The addition of *Lactobacillus acidophilus* in food has been correlated with higher levels of vitamin B12 and folate serum (postbiotic) (*p* < 0.05) and decreased anemia prevalence (*p* < 0.01) [48,68].

### 4.2. SCFA

SCFAs are the results of the fermentation of dietary fiber, resistant starch, and nonstarch polysaccharides (NSPs) in the human gut [69,70]. The composition of SCFA is variable and depends on the microbiota and the availability of the substrate. There are three main SCFAs: acetic acid (AA), propionic acid (PA), and butyric acid (BA). SCFAs are produced in almost all parts of the intestine, but mainly in the proximal part of the large intestine. SCFAs as organic acids have the ability to lower intestinal pH. On the one hand, these actions are beneficial because they improve the bioavailability of metals and create a protective barrier against colonization by pathogenic microorganisms [71]. On the other hand, the acidification of the intestinal tract may damage intestinal cells [72].

SCFAs, such as butyrate, propionate, and acetate have been recognized as mediators of iron absorption [73,74]. Data on the relation between SCFAs and anemia in animal studies are contradictory. Soriano et al., showed an increase in the concentration of all SCFAs in mice with anemia [75], but in Dostal et al., a decrease in intestinal SCFA-producing bacteria, such as *Roseburia* spp., *Eubacterium rectale* group was found in iron-deficient rats; propionate and butyrate were also diminished, thus there may be species differences. In infants with iron deficiency anemia, a reduction in the relative abundance of butyrate-producing species, such as Roseburia, Coprococcus, and Butyricicoccus, without a change in fecal butyrate levels was found [73]. Propionate and butyrate have been shown to increase hypoxia-induced factor 2α (HIF2α) RNAm and inhibit HIF2α activity in vitro [76]. HIF2α is a transcription factor that regulates iron absorption in the duodenum. HIF2α influences three key genes regulating iron absorption: ferroportin, divalent metal transporter 1 and duodenal cytochrome b [74]. In conclusion, an increase in SCFA during iron deficiency anemia seems to improve iron absorption.

### 4.3. Tryptophan

Tryptophan (Trp) is one of the eight exogenic amino acids that is used in the synthesis of proteins. Tryptophan metabolism in the human gut takes place through three main pathways: (1) direct metabolism by the intestinal microbiome, (2) kynurenine pathway in host cells, and (3) enzyme transformation to serotonin (5-HT). The intestinal metabolism of tryptophan regulates the total available tryptophan, influencing the serotonin system and the immune system. The kynurenine pathway, as the main pathway of tryptophan catabolism, is of increasing interest because its dysregulation is believed to be associated with inflammation, tumor proliferation, neurodegenerative diseases, and depression. The availability and metabolism of tryptophan have been reported to be associated with malabsorption (i.e., fructose malabsorption syndrome) and mood disorders. Increased activity of indoleamine 2,3-dioxygenase with enhanced tryptophan degradation is also considered to be involved in the drop in blood hemoglobin levels and the development of anemia [77,78,79]. In studies by Wenninger et al., decreased levels of tryptophan in the serum were correlated with lower hemoglobin values. At the same time, a 10% increase in tryptophan is associated with an increase in the level of hemoglobin by 1.59%. The availability of tryptophan is a limiting factor in protein biosynthesis and cell growth and is also important in erythropoiesis. Current data show that serum tryptophan levels correlated with ferritin iron reserve protein (*p* = 0.038, r = 0.100). A 10% increase in tryptophan is associated with an increase in ferritin of 6.45% [80]. The study found that patients with anemia had a lower level of tryptophan in the serum and an increased rate of tryptophan decomposition compared to patients without anemia [80].

The pro-inflammatory cytokines INF-gamma and TNF-alpha and the indoleamine 2,3-dioxygenase enzyme are involved in the degradation of tryptophan, which inhibits erythropoiesis [81]. As a consequence, the breakdown of tryptophan into kynurenine intensifies the inflammation, which results in a deficiency of this essential amino acid. At the same time, serum tryptophan tends to covalently bind to albumin. Since albumin is a negative acute-phase protein, it additionally lowers the tryptophan level [82,83].

Additionally, some authors reported that in probiotic-related fermented food or beverages, lactic fermentation also may have a positive effect on iron bioavailability. El-Azeem et al., in an animal model showed that probiotic-related fermented milk beverage increased iron absorption in rats [84]. Scheers et al., reported in human subjects that using lactic fermented vegetables with meals increased iron bioavailability. The authors used a culture of *L. plantarum* for vegetable fermentation which was later added to the meals. They found that lactic fermented vegetables added to the meal increased iron absorption in humans when compared to the fresh ones [85]. Similarly, Adeyanju et al., in their in vitro study used *L. plantarum* culture for the fermentation of sorghum and amaranth, and they found that non-alcoholic beverages made of these cereals have increased iron bioaccessibility [86]. The summarized studies with postbiotics are listed in Table 4.

## 5. Summary

Unfortunately, we currently do not have enough research that can provide in-depth information on the use of pre-, pro, and postbiotics in anemia, and their therapeutic or preventive effects on anemia. The most common IDA affects over 2 billion people worldwide. Surprisingly, IDA appears to be easily treatable with diet and medication, but the statistics above do not support this thesis. Hence, new ways to prevent and treat anemia and even support iron supplement treatment are desired. So far, it is known that probiotics, especially iron-enriched GOS, significantly improve the health, and iron status of patients. It is also known that the probiotic bacteria *Lactiplantibacillus plantarum 299v* in the gut improves iron absorption, preventing anemia. However, the recently completed Human Microbiome Project, although providing data on the impact of microbes on the host’s health, underlined the need for further research [87]. Postbiotics, such as SCFA have the most documented role in maintaining iron status. The latter act through acidification, but also mediate iron transport. In vitro acidification as a result of fermentation with probiotic bacteria also improves iron absorption. However, there is a lack of research on the parallel use of pre and probiotics, i.e., synbiotics. In conclusion, prebiotics, probiotics and postbiotics influence iron absorption and its condition in humans and should be considered in the holistic treatment of iron deficiency anemia.

## Figures and Tables

**Table 3 microorganisms-10-01330-t003:** Studies evaluating probiotics in anemia.

Study	Subjects	Intervention	Number of Patients	Duration of the Study	Outcome
Korcok D.J. et al., (2018) [25]	healthy humans, women	*L. plantarum* 299v 1.1 × 10^9^ CFU vs. placebo	20	1 week	significant increase in serum iron level in the probiotic group
Adiki S.K. et al., (2019) [59]	animal model, rats	*L. plantarum* 299v in 0.5 g dose and 1.0 g dose vs. different diet and iron supplementation	42, 7 groups per 6 rats	4 weeks	significant increase in iron absorption in lower probiotic group vs. diet groups, no differences in iron absorption between the group with higher and lower doses of probiotics
Hoppe M. et al., (2017) [60]	healthy humans, women	*L. plantarum* 299v 10 × 10^10^ CFU vs. placebo	14, study 1 28, study 2	4 weeks	significant increase in serum iron levels in probiotic groups compared to placebo groups in both studies
Garces V. et al., (2018) [61]	animal model, rats	*L. fermentum* alone, with or without iron oxide nanoparticles vs. iron supplementation	30, 5 groups per 6 rats	23 days	significant increase in the absorption of iron in probiotic with iron oxide nanoparticles group compared to probiotic alone and iron oxide nanoparticles alone
Khodaii Z. et al., (2019) [62]	animal model, rats	Fermented bread with or without *L. acidofilus*	24, 4 groups per six rats	30 days	significantly increased serum ferritin level and intestinal tissue cells compared to controls
Rosen G. et al., (2019) [64]	children with iron deficiency anemia	Iron supplementation with or without *L. plantarum* 299v 10 × 10^9^ CFU	52, 27 in the probiotic group, 25 in the placebo group	3 years	no significant differences in serum ferritin levels between the probiotic and control group

**Table 4 microorganisms-10-01330-t004:** Studies evaluating postbiotics in anemia.

Compound/Authors	Study/Control Group	Method	Effect
FA; Strozzi G.P. et al., (2008) [66]	23 healthy volunteers were randomly assigned to 1 of 3 study group	determination of the folate concentration in feces evacuated within 48 h before and after administration strains	significant increase in FA concentration in all treated groups (*p* = 0.004, *p* < 0.001, *p* = 0.049)
FA; Mohammad O. et al., (2009) [48]	12 children in the study and control group	analysis of folate plasma concentration after 42 days *Lactobacillus acidophilus* supplementations	increased folate plasma concentration (*p* < 0.01), reduction in the percentage prevalence of anaemia (*p* < 0.01)
SCFA; Soriano-Lerma A. et al., (2022) [75]	20 male Wistar rats, 11 in the control group, 9 in the anemic group	diet induction IDA for 40 days. Measured SCFA concentration in GI tract.	Significant increase in AA, PA and BA in the colon (*p* < 0.05) in rats with anemia
SCFA; Dostal A. et al., (2012) [76]	40 male Sprague-Dawley rats. Thirty-seven rats in the study group, 3 rats in the control group	3 rats on a normal diet and 37 rats on a non-iron diet for 24 d. After 37 d. cecal SCFA measurement	the cecal concentration of butyrate was 87% lower and that of propionate was 72% lower compared to the control group (*p* < 0.05).
Trp; Wenninger J. et al., (2019) [80]	115 patients with iron deficiency and 315 individuals without iron deficiency.	correlated between tryptophan and iron metabolism and hemoglobin levels in a large cohort of patients grouped by the presence or absence of iron deficiency or anemia.	indicators of tryptophan metabolism were positively correlated with haemoglobin and ferritin (*p* < 0.001; *p* = 0.038)
El-Azeem A. et al., (2016) [84]	45, 3 groups per 15 rats	*Streptococcus thermophilus* fermented milk beverage enriched diet vs a normal and tannic acid enriched diet	increase in serum hemoglobin, iron, and ferritin in probiotic fermented milk beverage enriched diet group compared to normal and tannic acid enriched diet groups
Scheer N. et al., (2016) [85]	17 rats	*L. plantarum* 299v 2.4 × 10^9^ CFU fermented vegetables vs fresh vegetables	significantly increase in iron absorption in the probiotic fermented vegetable group compared to the fresh vegetable group

## Data Availability

Not applicable.

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
