# Peer review of "Prebiotics, Probiotics, and Postbiotics in the Prevention and Treatment of Anemia"

_microorganisms, 2022, doi:10.3390/microorganisms10071330_

Round 1

Reviewer 1 Report

The aim and the novelty character of this review respect to those present in literature should be better marked.

The authors should describe the main features and advances od research of pre-. pro-, postbiotics and related references should be added such as:

Durazzo et al. An Updated Overview on Nanonutraceuticals: Focus on Nanoprebiotics and Nanoprobiotics. Int. J. Mol. Sci. 202021, 2285. https://doi.org/10.3390/ijms21072285

A section Methodology including bibliographic research criteria should be included.

Data in table 1 and 2 should be better discussed in the text

Author Response

Sir/Madam,

Thank you very much for your valuable comments.

The aim and the novelty character of this review respect to those present in literature should be better marked. The authors should describe the main features and advances od research of pre-. pro-, postbiotics and related references should be added such as: Durazzo et al. An Updated Overview on Nanonutraceuticals: Focus on Nanoprebiotics and Nanoprobiotics. Int. J. Mol. Sci. 202021, 2285. https://doi.org/10.3390/ijms21072285

Thank you especially for this paper, we included some data as well add this article to references.

A section Methodology including bibliographic research criteria should be included.

The paper is a review, not systematic review or metaanalysis, so in our opinion such a section is not necessary

Data in table 1 and 2 should be better discussed in the text.

Improved and the tables are better linked to the text.

Reviewer 2 Report

Dear Authors,

In general, I would consider the paper to be good.

The question is original, is well defined and is very relevant.

So it was very necessary to try to pull together all of this information and draw possible conclusions from it. This paper does that and the results provide an advance in current knowledge.

Very good search of the literature appears to have been carried out and that is a key issue here.

However, it would be interesting to provide additional explanations on the possible actions of Prebiotics, probiotics and postbiotics on the treatment and prevention of anemia. Use figures and graphs to better illustrate your explanations.
Also, it would be desirable if you could change the presentation of the tables which are difficult to read/understand.

Best regards,

Author Response

Sir/Madam

We are very grateful for your kind opinion and comments.

Dear Authors,

In general, I would consider the paper to be good.

The question is original, is well defined and is very relevant.

So it was very necessary to try to pull together all of this information and draw possible conclusions from it. This paper does that and the results provide an advance in current knowledge.

Very good search of the literature appears to have been carried out and that is a key issue here.

However, it would be interesting to provide additional explanations on the possible actions of Prebiotics, probiotics and postbiotics on the treatment and prevention of anemia. Use figures and graphs to better illustrate your explanations.

Yes, you are right, but these figures would be also complicated and make the manuscript longer. We will prepare a graphical abstract.  
Also, it would be desirable if you could change the presentation of the tables which are difficult to read/understand.

Tables 1 and 2 were shortened

Reviewer 3 Report

In this manuscript, the authors reviewed the function of prebiotics, probiotics, and postbiotics in the therapy of anemia. Overall, it updates the recent findings in this area.  It will be good to briefly discuss the correlation among these three or their correlation in anemia. In addition, some minors are suggested.

Line 37, e.g. should be such as, and other locations, such as line 39, line 40, …

Line 78, bacterial phyla names do not need to be italic.

Line 94, short-chain fatty acid should be abbreviated here, but not line 104.

Line 104, post-biotics > postbiotics, to be the same with others.

Line 125, Fe3+ > Fe3+, also check other chemical expression FeSO4 > FeSO4.

Table 1, Main Result, capitalize the sentence first word and list all the abbreviations including FePP.

Author Response

Thank you for your positive evaluation.

In this manuscript, the authors reviewed the function of prebiotics, probiotics, and postbiotics in the therapy of anemia. Overall, it updates the recent findings in this area.

 It will be good to briefly discuss the correlation among these three or their correlation in anemia.

Lack of such a data –such information was added in summary

In addition, some minors are suggested.

Line 37, e.g. should be such as, and other locations, such as line 39, line 40, …

This part due to other comments were deleted.

Line 78, bacterial phyla names do not need to be italic.

corrected

Line 94, short-chain fatty acid should be abbreviated here, but not line 104.

corrected

Line 104, post-biotics > postbiotics, to be the same with others.

corrected

Line 125, Fe3+ > Fe3+, also check other chemical expression FeSO4 > FeSO4.

corrected

Table 1, Main Result, capitalize the sentence first word and list all the abbreviations including FePP.

corrected

Reviewer 4 Report

The idea of this review is interesting, but the manuscript does not bring much novelty and it requires many clarifications.

1. Title: should refer to “iron deficiency anemia” not just generally “anemia”, since this is the main topic, the authors approached.

2. Abstract: a. According to the WHO, the definition of anemia involves only the decreased level of Hemoglobin. b. Please use the term “microbiota”, instead of “microflora”. c. Aim in the Abstract is different than the one in the main manuscript. d. The authors wrote “This review presents the actual data from research studies on the influence of probiotics, prebiotics, and postbiotics on the prevention and therapy of different types of anemia”…however, the main text is focused on iron deficiency anemia.

3. Introduction: a. Same comment about definition of anemia. b. Children are also often affected – line 35. Not mentioned. c. Among microcytic anemias – inflammatory anemia is not mentioned. d. Among normocytic anemia, leukemia and other infiltrative disorders of the bone marrow are not mentioned either and they are more frequent than myeloma. I see the references that the authors used ([1 and 2]), but they are not the most correct. d. Lines 49-50 – there are also other causes, more frequent of iron deficiency anemia – like occult bleeding – from the GI tract, gynecological etc; or high requirements during different periods of age. They were not mentioned. e. The first paragraph of Introduction presents aspects related only to iron deficiency anemia and emphasizes the iron roles. Nothing about other types of anemia.

4. Prebiotics – Since this review is about prebiotics and anemia, I suggest to remove history of definition and present the actual one (ISAPP 2017). Paradoxically, this reference of utmost importance lacks completely. I suggest the authors to read this paper and revise accordingly this paragraph.

5. Probiotics – also this paragraph is too long. Please present the last definition of probiotics and the most important genera. Besides, I do not think a subdivision in paragraphs is necessary, since each of them would contain only a few sentences. Again, surprisingly, the definition of probiotics by ISAPP (Hill C, et al, 2014) is missing. Please insert it and refer to it, also when presenting their mechanism of action. Please mention, very important, that the effect of a probiotic is strain specific and only in a given, established (by scientific evidence) dosage.

6. Postbiotics – this paragraph has to be entirely re-written, as the authors did not read and include the very recent definition of Postbiotics by ISAPP (Salminen et al, 2021). Strange how the authors wrote a review on pro-, pre- and postbiotics and were not aware that they have to refer to the ISAPP consensus manuscripts (The International Scientific Association of Probiotics and Prebiotics).

7. The aim of this review should be presented clearly. It appears by the end of paragraph Postbiotics. Moreover, it refers only to iron deficiency anemia, contrary to the abstract. Aims have to appear the same in the abstract and main manuscript.

8. Prebiotics in anemia: a. This paragraph is way too long. Too many sentences regarding each type of prebiotic. Please focus on the aim of the review. b. Also, main content is about iron and iron-deficiency anemia. The abstract mentions “different types of anemia”. Please revise. c. The authors wrote: Lines 180-181: “The summarized animal studies with prebiotics are listed in Table 1.”. However, title of Table 1 refers to ”human studies”, as well as its content. Please correct.

9. Probiotics in anemia: a. Please use the new name of Probiotics from genus Lactobacillus, like Lactiplantibacillus plantarum, Limosilactobacillus fermentum and so on (from ref. Zheng et al, 2020). This paper was published 2 years ago and the authors should have been aware about the change of the name of some probiotics.

10. Postbiotics in anemia: a. I suggest the authors to carefully revise this paragraph, according to the ISAPP definition of Postbiotics. b. Also, please reduce its content and focus on the main topic.

 11. Summary: This paragraph ends with a conclusion about iron deficiency anemia. Again, please revise the aim of the manuscript so that the conclusion fits the objective of the review.

Author Response

The authors highly appreciate the reviewer’s positive feedback and would like to thank for his/her comments that helped to improve the manuscript. In the following all points of the reviewer are addressed:

The idea of this review is interesting, but the manuscript does not bring much novelty and it requires many clarifications.

  1. Title: should refer to “iron deficiency anemia” not just generally “anemia”, since this is the main topic, the authors approached.

Yes, it is true that we mostly describe iron deficiency anemia, but not only, we are also discuss folic acid, so we improve the aim of the study, but we do not change the title. 

  1. Abstract: a. According to the WHO, the definition of anemia involves only the decreased level of Hemoglobin.

corrected

  1. Please use the term “microbiota”, instead of “microflora”.

corrected

  1. Aim in the Abstract is different than the one in the main manuscript.

Corrected

  1. The authors wrote “This review presents the actual data from research studies on the influence of probiotics, prebiotics, and postbiotics on the prevention and therapy of different types of anemia”…however, the main text is focused on iron deficiency anemia.

Yes, you are right, this problem in mention in other points. We mostly describe iron deficiency anemia, but not only, we are also discuss folic acid, so we improve the aim of the study in abstract as well in the text body.

  1. Introduction: a. Same comment about definition of anemia. b. Children are also often affected – line 35. Not mentioned. c. Among microcytic anemias – inflammatory anemia is not mentioned. d. Among normocytic anemia, leukemia and other infiltrative disorders of the bone marrow are not mentioned either and they are more frequent than myeloma. I see the references that the authors used ([1 and 2]), but they are not the most correct. d. Lines 49-50 – there are also other causes, more frequent of iron deficiency anemia – like occult bleeding – from the GI tract, gynecological etc; or high requirements during different periods of age. They were not mentioned. e. The first paragraph of Introduction presents aspects related only to iron deficiency anemia and emphasizes the iron roles. Nothing about other types of anemia.

Thank you for these comments. This part of introduction was recomposed and improved as recommended. Description of anemia types was deleted as not necessary.

  1. Prebiotics – Since this review is about prebiotics and anemia, I suggest to remove history of definition and present the actual one (ISAPP 2017). Paradoxically, this reference of utmost importance lacks completely. I suggest the authors to read this paper and revise accordingly this paragraph.

Corrected, proper reference added

  1. Probiotics – also this paragraph is too long. Please present the last definition of probiotics and the most important genera. Besides, I do not think a subdivision in paragraphs is necessary, since each of them would contain only a few sentences. Again, surprisingly, the definition of probiotics by ISAPP (Hill C, et al, 2014) is missing. Please insert it and refer to it, also when presenting their mechanism of action. Please mention, very important, that the effect of a probiotic is strain specific and only in a given, established (by scientific evidence) dosage.

Corrected according to suggestions, definition and reference added

  1. Postbiotics – this paragraph has to be entirely re-written, as the authors did not read and include the very recent definition of Postbiotics by ISAPP (Salminen et al, 2021). Strange how the authors wrote a review on pro-, pre- and postbiotics and were not aware that they have to refer to the ISAPP consensus manuscripts (The International Scientific Association of Probiotics and Prebiotics).

Corrected, indicted definition was added.

With all the respect to great work of ISAPP, there are many others very significant organizations like WHO or WGO, which do not implement definitions suggested by ISAPP.

  1. The aim of this review should be presented clearly. It appears by the end of paragraph Postbiotics. Moreover, it refers only to iron deficiency anemia, contrary to the abstract. Aims have to appear the same in the abstract and main manuscript.

Corrected, explanation above

  1. Prebiotics in anemia: This paragraph is way too long. Too many sentences regarding each type of prebiotic. Please focus on the aim of the review. b. Also, main content is about iron and iron-deficiency anemia. The abstract mentions “different types of anemia”. Please revise. c. The authors wrote: Lines 180-181: “The summarized animal studies with prebiotics are listed in Table 1.”. However, title of Table 1 refers to ”human studies”, as well as its content. Please correct.

Paragraph was shortened, and corrected according to suggestions

  1. Probiotics in anemia: a. Please use the new name of Probiotics from genus Lactobacillus, like Lactiplantibacillus plantarum, Limosilactobacillus fermentum and so on (from ref. Zheng et al, 2020). This paper was published 2 years ago and the authors should have been aware about the change of the name of some probiotics.

corrected according to suggestions, however old taxonomy is still common in scientific papers

  1. Postbiotics in anemia: a. I suggest the authors to carefully revise this paragraph, according to the ISAPP definition of Postbiotics. b. Also, please reduce its content and focus on the main topic.

The paragraph was shortened. We do not agree completely with definition of postbiotics proposed by ISAPP. We are not alone, we think, because some additional explanation was delivered by ISAPP in Vinderola`s paper this year. However, our manuscript did not discuss with the ISAPP definition.

  1. Summary: This paragraph ends with a conclusion about iron deficiency anemia. Again, please revise the aim of the manuscript so that the conclusion fits the objective of the review.

Summary was revised and composed de novo.

Round 2

Reviewer 4 Report

I highly appreciate the improvements the authors made in their review. I support the publication of the manuscript. Just some very minor modifications are needed:

Line 132: Please delete “Kliknij lub naciÅ›nij tutaj, aby wprowadzić tekst.”(text in Polish)

Line 367: Please correct to “Lactiplantibacillus”, in order to ensure uniformity.

Author Response

Dear Reviewer,

Thank you for your positive evaluation.

Line 132: Please delete “Kliknij lub naciÅ›nij tutaj, aby wprowadzić tekst.”(text in Polish)

Corrected

Line 367: Please correct to “Lactiplantibacillus”, in order to ensure uniformity.

Corrected